# Point-of-care ultrasound use in austere environments: A scoping review

**Aubree Anderson[1], Rebecca G. Theophanous[1,2]***

**1** Department of Emergency Medicine, Duke University School of Medicine, Durham, NC, United States of America, **2** Durham Veterans Affairs Healthcare System, Durham, NC, United States of America

\* Rebecca.theophanous@duke.edu

**Data Availability Statement:** All relevant data are within the manuscript and its Supporting Information files.

## Abstract

### Background/Objectives

Technological developments in point-of-care ultrasound (POCUS), particularly with portable devices, are transforming POCUS use in austere, resource-limited environments (RLS) distinct from typical hospital or medical settings. POCUS has potential to improve diagnostic accuracy in military combat zones, low-resource environments such as the desert or tropics, microgravity, and high altitudes. Our updated narrative scoping review describes POCUS use in these global settings.

### Methods

Using the PRISMA-ScR guidelines, two ultrasound-trained emergency physicians searched PubMed, Embase, and Web of Science on August 6, 2024 for "point-of-care ultrasound in austere environments" and each individual category. Study titles and abstracts were independently screened, then full manuscripts, and data was abstracted with a data collection table. 324 articles met inclusion criteria: research studies describing POCUS in austere environments; involving healthcare professionals; and in English. We excluded abstracts, studies not involving POCUS in austere environments, and non-clinical studies. Reviewers critically appraised studies using the GRADE (Grading of Recommendations, Assessment, Development, and Evaluations) Quality Assessment Tool.

### Results

There were 39 military or conflict zone studies, 101 prehospital, 148 in RLS including low- and middle-income countries, 12 outer space, 15 high altitude, and 32 involving POCUS use in multiple austere environments. There were 6 randomized-control trials, 11 systematic/scoping reviews, 13 narrative reviews, 112 prospective observational/cohort, 34 prospective cross-sectional studies, 23 retrospective, 6 feasibility, 45 case reports, 13 case series, and 5 educational curriculum studies. GRADE study quality was variable, with 74 high quality, 129 moderate, 82 low, and 56 very low.

### Conclusion

The existing literature is mixed with variability in study settings, design, and POCUS examination types, providing an initial understanding of POCUS applications. Most studies are in

**Funding:** The author(s) received no specific funding for this work.

**Competing interests:** The authors have declared that no competing interests exist.

RLS or prehospital settings. Additional high-quality studies are needed to guide POCUS training, disseminate use in non-hospital settings, and maximize impact for improved clinical outcomes in diverse austere environments.

## Introduction

Point-of-care ultrasound use (POCUS) in austere environments is an exciting and developing topic, especially with rapid improvements in technology and miniaturization of POCUS over the past 15–20 years [1, 2]. Austere environments can be defined as locations that are outside of the standard hospital or clinic setting, sometimes with extreme temperatures, remote locations, or unique settings such as in deserts or outer space. These environments create challenges to using POCUS, including physical machine deterioration from hot or cold temperatures, high humidity, and exposures such as rain or sand. Additional problems when using POCUS outside of the typical medical setting include battery degradation, failure of hard drives, poor or no wireless connectivity, variable ambient sunlight intensity, and often limited immediate or urgent technological support for equipment [1, 2]. Due to these additional limitations in operating equipment in these unique and sometimes remote locations, POCUS machines must be lightweight, durable, easy to operate, and have adequate battery life to facilitate increased diagnostic accuracy in the field [1, 2].

Since 2009, the invention of portable POCUS devices has revolutionized the emergency medicine and ultrasound field [3]. The potential utility of POCUS is broad, facilitating views of deep organs or superficial tissues to assist in diagnostic and treatment decisions within minutes. POCUS use has been described in diverse settings, including in outer space, deserts, jungles, mountains, the sea, and more [1–3]. Its portability, safety with no ionizing radiation, live images with rapid feedback, and potential for image transmissibility makes it an ideal tool for use in limited resource settings for both diagnostic and procedural applications [1, 3, 4].

Studies describe POCUS use in military medicine, outer space, high altitudes, and other resource-limited settings (RLS), including in low and middle-income countries (LMIC) [1, 2]. For example, POCUS can be built into a simple triage and rapid treatment or "START" triage algorithm for natural disasters such as earthquakes and floods or in mass casualty situations including combat zones [3, 5, 6]. Using POCUS to identify intra-abdominal or pericardial free fluid, pneumothorax, or cardiac activity helps assign patients to green, yellow, red, or black categories for mobilization and allocation of limited treatment resources [1, 5]. With growing portable POCUS capabilities using more lightweight and smaller machines with preserved image quality, POCUS expansion and adoption into a multitude of environments has potential to improve patient care and clinical outcomes. This article delves into the literature and highlights current and potential applications for POCUS in austere environments. The study aim was to collect data from an updated literature review on current POCUS use in austere environments to help understand current existing barriers to care and identify potential opportunities for future development and expansion.

## Methods

We performed a scoping review on POCUS in austere settings given the breadth of the research topic, variation in study design, and study heterogeneity across settings, applications, devices, and users. The study was performed using the Joanna Briggs Institute's (JBI) approach and following PRISMA scoping review guidelines. The study did not require informed consent

or Institutional Review Board approval (**S1 Checklist**). PROSPERO does not allow registration of scoping reviews thus no online review protocol exists.

## Literature search strategy

With medical librarian assistance, PubMed, Embase, and Web of Science were systematically searched on August 6, 2024 for studies in English using the search terms "point-of-care ultrasound in austere environments" and for each study setting category of *1) military and conflict zones*, *2) prehospital (including emergency medical services or EMS)*, *3) RLS including LMIC*, *4) microgravity in outer space*, *and 5) high altitude*. We considered publications that were full manuscripts, published in peer-reviewed journals, and in English. Studies were included with any study design, from all countries, and no date limits were set. The search criteria were maintained broad to capture the current existing literature base for POCUS in austere environments as is appropriate for a scoping review. Studies were then screened and descriptively analyzed, by reviewers for comparisons. The full search criteria and terms are included in **S1 Appendix**. All citations were imported into a comprehensive library using Endnote version 20.6 (Clarivate, Philadelphia, PA, USA) and deduplicated, yielding 766 articles.

## Study selection

The Population/Concept/Context (PCC) framework was used to help create a clear title and study question regarding POCUS use in austere environments and to inform the inclusion criteria. Inclusion criteria were: published full research manuscripts; describing POCUS in austere environments; involving healthcare professionals (e.g. physicians, prehospital medics, mid-level or other area-specific local healthcare providers); from any publication year; and in English. Studies were excluded if they were not full publications or primary research literature, did not focus on POCUS in austere environments or RLS, or were not in English.

## Study screening

Two ultrasound-trained emergency medicine physicians (AA and RT) blinded to each other independently screened article titles and abstracts for study inclusion. Disagreements were resolved with discussion on the second round, with minimal changes required. Articles were re-screened by abstract then by full text using the same methods. Authors for studies with abstracts only were contacted via email on August 10, 2024 by RT to request full manuscripts for screening and review without additional papers obtained.

## Data abstraction

Data was extracted independently by each reviewer using a standardized data collection tool on an Excel spreadsheet (Microsoft Corporation, Redmond, WA, USA, version 2408) and input into a summative table. Studies were critically appraised for study quality and risk of bias using the GRADE (Grading of Recommendations, Assessment, Development, and Evaluations) NIH Quality Assessment Tool [7].

## Data charting and collation

Data was charted using an Excel spreadsheet and included: author name and publication year, study design, study topic (POCUS examination type), and study setting (S1A–S1F Table). Missing data on ultrasound devices or technologies was designated as "not reported" (NR) in the table. Data was collated from descriptive analysis and discussion between the two reviewers and organized by 1) Military medicine and conflict zones, 2) Prehospital (including

Emergency Medical Services), 3) RLS including LMIC, 4) Microgravity in outer space, and 5) High altitude and mountains. Findings were then synthesized into common themes for comparison across study settings. Major trends were identified for types of POCUS examinations and current applications utilized in austere environments. The findings can then help guide future implementation projects to facilitate POCUS use and address barriers in these settings.

## Results

### Study characteristics

The initial search generated 1159 articles from all three databases (1121 PubMed, 31 Embase, 7 Web of Science), with 393 duplicate studies removed. We excluded 424 studies based on title and abstract and 18 excluded due to no abstract or full paper. The remaining 324 articles were extracted for full-manuscript analysis and evaluated for eligibility criteria. These were then organized into categories based on study setting (**Fig 1** and **S1A–S1F Table**: Summary of included ultrasound in austere environments studies in narrative scoping review).

Table 1 lists the summary descriptive data for the included studies. Of the 324 articles that met eligibility criteria, there were 39 studies in military or conflict settings, 101 prehospital (emergency medical services), 148 studies in RLS including LMIC, 12 microgravity in outer space, 15 high altitude, and 32 involving POCUS use in multiple austere environments (**S1A–S1F Table**). There were 6 randomized-control trials, 11 systematic/scoping reviews, 13

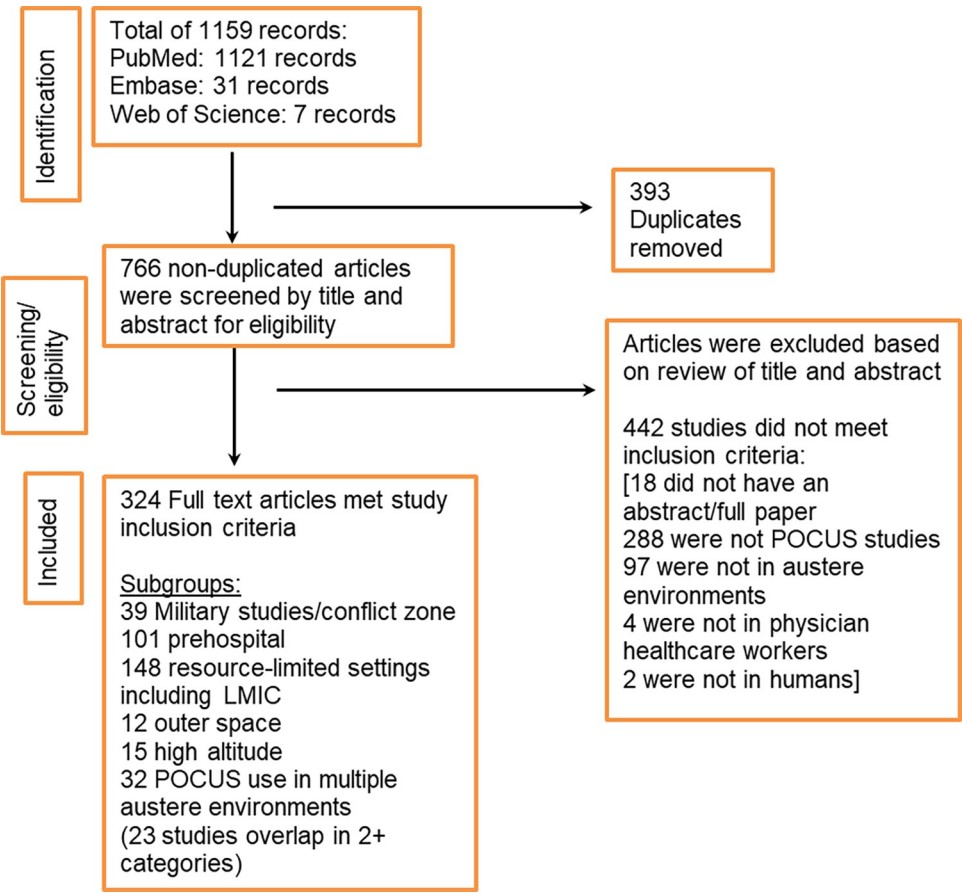

**Fig 1. Flow diagram of austere environment ultrasound study screening and selection.**

**Table 1. Summary descriptive data of ultrasound in austere environments studies included in the narrative scoping review.**

| Item | Article and study types |
|---|---|
| Total studies from literature search (PubMed, Embase, Web of Science) | 1159 articles (393 duplicates removed) thus 766 studies remaining |
| Excluded studies | 442 studies |
| Included studies | 324 studies |
| Number of studies included by setting: | 39 military/conflict zone |
| | 101 prehospital/EMS |
| | 148 resource-limited settings including LMIC (11 TB/FASH/HIV studies, 6 Lung/COVID-19 studies, 4 acute heart failure/cardiac studies, 1 handheld African ED, 7 handheld cardiac studies, and 25 POCUS curriculum evaluation studies) |
| | 12 outer space |
| | 15 high altitude |
| | 32 POCUS use in multiple austere environments |
| | (23 studies overlap in 2+ categories) |
| Number of included studies by design: | 6 randomized-control trials |
| | 11 systematic/scoping reviews |
| | 13 narrative reviews |
| | 112 prospective observational/cohort |
| | 34 prospective cross-sectional studies |
| | 23 retrospective |
| | 6 feasibility |
| | 45 case reports |
| | 13 case series |
| | 5 educational curriculum studies |
| Number of included studies by quality of evidence (GRADE assessment tool): | 74 high quality |
| | 129 moderate |
| | 82 low |
| | 56 very low |

narrative reviews, 112 prospective observational/cohort, 34 prospective cross-sectional studies, 23 retrospective, 6 feasibility, 45 case reports, 13 case series, and 5 educational curriculum studies (**Table** 1). Study quality was variable, with 74 high quality, 129 moderate, 82 low, and 56 very low per the GRADE assessment tool (**Table** 1) [7]. **S1A–S1F Table** lists the detailed study characteristics and quality of evidence for each study, grouped by category. **S2 Table** lists a summary of the prehospital and RLS including LMIC studies for additional comparison due to the higher number of studies found in those categories. **S3 Table** is the total numbered list of all included studies and **S4 Table** is the total numbered list of excluded studies with reasons for exclusion.

## Main results

We present a descriptive overview of the study results below.

**A) Military medicine and conflict zones.** The number of military and conflict zone studies was low (39 studies), with regions including the United Kingdom, United States, France, Israel, Iraq, Australia, and China. United States military base locations include in Tacoma, WA; San Antonio, TX; Seattle, WA; and Fort Hood, TX. Study types were primarily low to moderate quality, including case reports, cross-sectional studies, prospective cohort, and

retrospective. A few narrative reviews described multiple examination types such as extended focused assessment with sonography in trauma (EFAST), musculoskeletal (MSK) for fractures, soft tissue for foreign bodies or abscess, procedural nerve blocks, and lung ultrasound [8–10]. Some studies were older, in the early 2000s, ranging to more recent studies within the past 5 years.

**B) Pre-hospital medicine.** The prehospital literature search found 101 studies in numerous countries (e.g. United States, England, Israel, Portugal, Laos, etc.). There were a mix of prospective and retrospective studies and reviews. Some RCTs were done, including one by Chen et al evaluating EMS providers with and without teleultrasound in Israel in 2022 [11]. Cardiac and lung ultrasounds were prevalent, including in cardiac arrest patients to evaluate for pericardial effusion and in trauma patients to evaluate for pneumothorax [12, 13]. Other studies evaluated EFAST and aorta exams. Also, procedural POCUS use for vascular access, endotracheal intubation and gastric tube confirmation, lung sliding evaluation for needle thoracostomy, and pericardiocentesis in cardiac arrest. Some studies involved flight medicine [14].

**C) Resource-limited settings including low- and middle-income countries.** We reviewed 148 studies on POCUS in RLS including in LMIC. Reported diseases and settings significantly varied across all populated continents, including in Africa, Asia, Australia, Central and South America, Europe, Canada, and medical mission trips from the United States (**S1A–S1F Table**). Studies included both pediatric and adult patients, with case reports on POCUS use in tropical diseases including in extrapulmonary tuberculosis, echinococcus, malaria, liver abscess, purulent pericarditis, intussusception, congenital cardiac defects, and others [15–18]. Most studies were prospective, with smaller categories including POCUS training and needs assessment in LMIC, the Focused Assessment with Sonography for HIV/TB (FASH) exam, ocular US of optic nerve sheath diameter (ONSD) and splenic ultrasound for malaria, cardiac ultrasound for congenital and structural heart failure diagnoses, and antenatal and obstetric care [15–18].

**D) Microgravity in outer space.** Few POCUS studies in microgravity exist, with only 12 found in our review. Two studies are cross-sectional: simulated microgravity for parabolic flight of pneumothorax in pig models and doppler ultrasound for venous gas emboli and decompression illness in a hypobaric chamber simulation [19, 20]. Prospective cohort studies frequently incorporated "just-in-time training" of astronauts with some training preflight, and astronauts followed explicit written instructions while in spaceflight to acquire and interpret POCUS images. Examinations included internal jugular flow in different positions, spinal ultrasound, cardiac, and MSK shoulder ultrasound [21–24]. Only one study was a systematic review on lung ultrasound [25]. One scoping review by Asachi et al described multiple microgravity applications (abdominal, lung, deep veins, sinusitis, MSK, renal, ocular, and decompression sickness) [26]. Finally, three narrative reviews discussed the EFAST and teleultrasound use, cardiac and spinal ultrasound with physiological changes and decompression sickness, and a proposed POCUS curriculum for astronauts [27–29].

**E) High altitude and mountains.** For high altitude, we again found few studies (15 studies), with most being low quality case reports or case series, and a few small prospective cohort studies or reviews. Common topics included lung ultrasound for evaluation of high-altitude pulmonary edema (HAPE) or ocular ultrasound for ONSD [30–32]. Additional applications were soft tissue ultrasound for foreign bodies and MSK for fractures [33].

**F) POCUS use in multiple austere environments.** In general, POCUS use in austere environments is summarized in narrative reviews, with a few prospective studies and cross-sectional surveys. Many studies are a needs assessment for future planning. Case reports and series are described using ocular ultrasound for retinal detachment from a gunshot wound,

**Table 2. Most common point-of-care ultrasound applications in austere environments.**

| Austere and Extreme Environments | | | | |
|---|---|---|---|---|
| Military medicine and conflict zones | Prehospital (including Emergency Medical Services) | Resource-limited settings including low- and middle-income countries | Microgravity in outer space | High altitude and mountains |
| Focused assessment for sonography in trauma (FAST) | Lung (pneumothorax, hemothorax) | Abdominal (obstetrics, biliary, bowel/appendix, FAST, etc.) | Abdominal (biliary, appendix/bowel, obstetrics, etc.) | High altitude pulmonary edema (HAPE), pneumothorax |
| Lung (pneumothorax, hemothorax) | Cardiac (pericardial effusion, standstill in cardiac arrest) | FASH exam for tuberculosis (pericardial/pleural effusion, ascites, abdominal lymph nodes, splenic/liver lesions) | Nephrolithiasis, bladder | Musculoskeletal (fractures, tendons) |
| Cardiac | Abdominal (FAST, aorta) | Splenomegaly, ONSD in malaria | Deep venous thrombosis | Soft tissue |
| Soft tissue/MSK (foreign body, abscess, tendons, joints) | fractures | Cardiac (congenital or acquired structural disease, wall-motion abnormality, right ventricular dilation, etc.) | Musculoskeletal (tendons) | Optic nerve sheath diameter (ONSD) |
| Procedural (nerve blocks, vascular access) | Procedural (vascular access, gastric tube, needle thoracostomy, pericardiocentesis) | Lung (B lines, effusion, consolidation) | Spinal (disc herniation) | |
| | | Volume assessment of IVC | Ocular (Corneal abrasion, retinal detachment) | |
| | | Soft tissue (abscess, foreign body) | Cardiac | |
| | | Procedural | Lung | |
| | | | Procedural | |

foreign body removal using wilderness medical kits, and locoregional nerve blocks in caving accidents rescue [33–35]. Two case studies describe technological innovations for ultrasound dissemination, including remotely piloted aerial systems for drone delivery of a lung teleultrasound device and a smartphone video-based app for POCUS in RLS [36, 37]. High-quality literature is sparse, with one study by Volpicelli et al in 2012 creating expert consensus guidelines for lung ultrasound using the Delphi technique from three large conferences in Italy [38]. In 2023 Kaminecki et al performed a systematic review of POCUS for dehydration in children [39]. Finally, Maw et al performed a high-quality framework-based qualitative study using semi-structured interviews to evaluate POCUS program implementation [40].

Table 2 synthesizes data into the most common POCUS applications in austere environments and compares them side-by-side per study setting. Cardiac, lung, and intra-abdominal applications were used in all settings except high altitude for detection of internal organ injuries or pathology. Military and prehospital medicine were very similar in POCUS uses, with the most common applications being FAST for free fluid signifying internal hemorrhage, lung for pneumothorax assessment, and cardiac for pericardial effusion or cardiac standstill. Sometimes they performed MSK/soft tissue POCUS to evaluate for battle wounds from traumatic injuries such as gunshots, shrapnel, foreign bodies, or blunt injuries. High altitude had several unique POCUS applications for HAPE (identifying B lines in pulmonary edema) and ONSD (for increased intracranial pressure evaluation). RLS including LMIC was the most variable in POCUS examinations, likely due to variety in study settings and location needs or resources. Microgravity POCUS use was also broad as the primary imaging modality for astronauts in a confined space, identifying both medical (abdominal organ, nephrolithiasis, DVT, and cardiopulmonary physiological changes from microgravity), and traumatic injuries (spinal disc herniation, MSK, ocular corneal abrasion, procedural, etc.) By outlining the scoping review findings, the study aim was to help guide future implementation projects for facilitating POCUS expansion and overcoming barriers to current use in these settings.

## Discussion

The nature of extreme environments can lead to injuries and a need for rapid POCUS diagnostics in the field [1, 3, 8]. We provided an updated, detailed overview of the most common POCUS findings and uses in austere environments from an evidence-based scoping literature review.

### A) Military medicine, conflict zones, and B) pre-hospital medicine

We discuss military and pre-hospital POCUS use together as they contain major similarities, with both typically involving time-sensitive POCUS use in high pressure field environments [3, 4, 8]. The lower number of military and conflict zone studies (39 studies) may partly be due to dissemination in other venues not readily detectable in an online scientific literature search or required confidentiality for military operations data. Furthermore, it is difficult to perform research in natural disaster zones, which are unplanned natural events in usually low-resource settings, often with infrastructure instability requiring evacuations [2, 3]. The prehospital literature search was more robust, with 101 studies in many countries (e.g. United States, England, Israel, Portugal, Laos, etc.) (**S1A–S1F and S2 Tables**).

Point-of-care ultrasound is critical for military and pre-hospital medicine, who are usually in remote areas without access to advanced medical therapeutics. Medics must make quick decisions for field treatment versus mobilization or "scoop and run" expedited hospital transport [3, 4, 8]. Ultrasound machines have been made durable, with studies in arid desert climates or humid jungles showing device functionality and clear image transmission if satellite signals are available [1, 2, 4]. For example, for military or EMS use, performing an EFAST exam can detect intra-abdominal free fluid in a trauma victim, with high specificity to rule in a positive diagnosis (99.7%) [1, 4, 6, 9, 10]. Likewise, using lung ultrasound to detect absence of lung sliding, especially when a lung point is seen, is highly specific in detecting the presence of a pneumothorax prior to performing needle decompression [1, 6, 10]. (A lung point is defined as sliding of one part of the lung immediately adjacent to non-sliding lung in a single POCUS view). Furthermore, POCUS is useful in cardiac arrest to diagnose reversible pathology such as a large pericardial effusion causing tamponade, which can guide field medic treatment and mobilization decisions [1, 5, 6].

Studies with more recent technology can also incorporate remote guidance from an ultrasound expert via telehealth [1, 5, 6]. For example, POCUS identification of cardiac standstill could potentially assist in ending field resuscitative efforts by paramedics consulting with EMS medical directors via teleultrasound capabilities [1, 5, 6]. Other common military and conflict zone POCUS applications include soft tissue POCUS to detect foreign bodies, MSK to detect and stabilize fractures, and ocular trauma diagnosis including corneal defects or retinal detachments for expedited treatment [1, 9, 10].

### B) Resource-limited settings including low- and middle-income countries

The most studies pertaining to POCUS were found in LMIC (148 studies). Resources and medical personnel can be scarce in RLS, including in some LMIC, with many remote locations lacking a nearby clinic or hospital for medical care. Unique barriers exist in these settings. Patients sometimes must travel long distances to be evaluated and treated, and lack of transportation or funds can be additional barriers [1, 41–43]. Due to stigma related to disease diagnosis, lack of confidence with the medical system, and certain cultural customs that may prefer traditional medical treatments, people may avoid seeking medical care [44]. Furthermore, the available preventative healthcare system can be limited depending on local infrastructure, with low physician-to-patient ratios when patients live far from medical centers, inadequate

healthcare personnel training opportunities, and lack of governmental or financial support [41–43]. Tropical diseases can spread rapidly from poor sanitation and plumbing infrastructures in villages, lack of hygiene, and congregated living conditions in impoverished communities [45, 46]. Extreme conditions including food or crop scarcity in areas affected by natural disasters, such as drought, flooding, or storms, can exacerbate disease spread [45, 47]. Mosquito-borne illnesses such as malaria and water- or air-borne vectors for giardia or tuberculosis, for example, are also prevalent in certain geographic regions [15, 45, 48, 49].

Pathology requiring treatment can exist in all geographic regions, with common POCUS indications including trauma, cardiopulmonary, abdominal, and obstetrics/gynecology [42, 50–52]. POCUS can be used to help local medical personnel evaluate and treat specific medical conditions, for example detecting pleural or pericardial effusions, ascites, abdominal lymphadenopathy, and liver or splenic organ infiltration (FASH exam) in human immunodeficiency virus (HIV)-positive patients with tuberculosis [15, 53–55]. Also, splenomegaly or increased ONSD can be detected using POCUS in patients with malaria [16, 48, 56]. Other POCUS applications include diagnosing congenital or acquired structural heart disease, heart failure, deep venous thrombosis, obstetric complications, and volume assessment [17, 18, 57, 58]. A study in rural Ghana found that 71% of 67 POCUS scans performed in one month were abnormal, detecting a breast neoplasm, biloma, intrauterine fetal demise, ascites, and more [13]. The high percentage may be biased by only performing studies in patients with presumed abnormality, but the study highlights a broad spectrum of potentially detectable disease [59]. Another study done in the Amazon jungle showed that unnecessary patient transport was avoided in 28% of patients when POCUS was performed, ruling out gallstones and ectopic pregnancy to prevent costly and difficult transport to a higher level of medical care hours away [60]. POCUS also improved bedside diagnostic certainty by 72% [1, 60].

Finally, studies in POCUS curriculum development and training medical personnel on-site as ultrasound champions to teach others have shown success [61–64]. In a study in 2019, Nadimpalli et al demonstrated feasibility in training mid-level clinical officers (CO) to perform a POCUS algorithm for pediatric lung ultrasound in South Sudan. Of 360 POCUS scans, 99.1% of images were rated acceptable and 85% of CO interpretations were classified as appropriate per reviewers. They detected an excellent "inter-rater agreement between COs and experts for lung consolidation with air bronchograms kappa of 0.73 (0.63–0.82) and for viral lower respiratory tract infections/bronchiolitis kappa of 0.81 (0.74–0.87)" [65]. In 2020 Sabatino et al assessed the implementation of a POCUS training program for community health officers (CHOs) on cardiac, lung, and abdominal ultrasound using the EFAST exam in Lokomasama, a chiefdom of Sierra Leone. POCUS changed the initial diagnosis in 17% of cases. Learners achieved EFAST and POCUS knowledge scores of 90% and 83% post-training, with excellent inter-observer agreement (kappa 0.88) between CHOs and physicians [66]. Finally, Burleson et al created a POCUS fellowship called "PURLS" in 2020, which is an 18-month curriculum integrating academic clinical care and working in RLS to teach graduates the skills necessary for teaching contextualized ultrasound skills in RLS [67].

## C) Microgravity in outer space

POCUS is the primary imaging modality in outer space, as a light weight, portable, and easy to operate platform. Fewer POCUS studies exist in this field, with only 12 found in our review, with most being cross-sectional studies or narrative reviews. Nevertheless, studies have shown POCUS's fidelity and feasibility in microgravity, which causes changes in the human body such as muscle atrophy, bone demineralization, and cardiovascular deconditioning [26, 28]. The NASA handbook developed for astronauts in 2006 includes POCUS as one of the core

competencies in astronaut training to detect acute medical emergencies, including 4 hours dedicated to POCUS during mission training while on Earth [68]. At the International Space Station or while in flight, astronauts are trained to acquire ultrasound images and can transfer these down to Earth for expert review and guidance by ground medical staff [2, 3]. In 2003, as part of the Advanced Diagnostic Ultrasound in Microgravity (ADUM) program during the NASA Expedition 8 mission, astronauts successful performed diagnostic POCUS on themselves and their crewmates using real-time remote assistance from mission control [14]. Furthermore, due to time delays with travel at far distances in space, astronauts employ "just-in-time" training tactics for POCUS and other on-board mission tasks, by reviewing an ultrasound video tutorial then using the device and recording images [26, 69].

Asachi et al performed a scoping review of the most common exams performed in outer space, which include abdominal emergencies, decompression sickness (DCS), DVT, lung pathologies, MSK trauma, renal nephrolithiasis, and ocular corneal abrasion or retinal detachment [26]. They found that a FAST exam in a porcine model still detected free fluid in Morrison's pouch as the most sensitive area despite weightlessness in space, although additional studies are needed to confirm this finding [26]. Hamilton et al found that hemothorax fluid distribution in simulated porcine models is redistributed in space due to lack of gravity [19]. POCUS can also be used to diagnose venous air embolism for DCS [23]. In another study, POCUS was used to diagnose an internal jugular DVT to start anticoagulation on a crewmember at the ISS [20]. Finally, MSK POCUS can evaluate for tendon, muscle, or vertebral disc injuries, as the risk of rotator cuff tears and disc herniation is increased in astronauts in microgravity [21, 24]. Another study by Fischetti et al proposes a competency-based curriculum for astronauts including cardiac, lung, abdominal (biliary, bowel), renal (hydronephrosis, bladder), ocular, vascular (DVT), soft tissues, and procedural, which needs validation in future studies [29].

## D) High altitude and mountains

Finally, studies show that POCUS use in high altitude settings can help detect pulmonary edema, increased intracranial pressure, MSK injuries, and soft tissue foreign bodies and infections [31, 32, 71]. Hikers and climbers face acute mountain sickness and pulmonary edema due to low oxygen at high altitudes [31, 32, 60, 71]. POCUS can be used to detect lung B lines as an indicator of high altitude pulmonary edema (HAPE) in hikers or climbers, expediting treatment with descent and oxygen therapy [1, 8, 31, 32]. Two other field studies found that increased ONSD on ocular POCUS (suggesting increased intracranial pressure) correlated with increased acute mountain sickness scores [31, 32]. Hikers and climbers can also obtain MSK injuries from walking, ascending or descending, or maneuvering around obstacles such as rocks, branches, mud, sand, or unsteady terrain [31, 32, 71]. In one study of 20 non-ultrasound trained paramedics, MSK POCUS was used to detect long-bone fractures in less than 5 minutes, with a sensitivity of 97.5% (95% CI 94.1–100, $p<0.05$) and specificity of 95% (95% CI 85.4–100, $p<0.05$) [70, 71]. POCUS has also been used to diagnose abscesses for incision and drainage and to guide foreign body removal [1]. Another case report in 2001 described ruling out pneumothorax using POCUS in an 18-year-old skier with a blunt chest injury from a fall [72]. Finally, POCUS can be used to evaluate the lungs for pneumothorax prior to helicopter transports.

**Comparison of POCUS use across multiple resource-limited austere settings.** In summary, austere environments create a unique setting for POCUS use that exhibit both similarities and differences. For example, studies done in Iraq describe military personnel treating trauma patients in combat zones in dry, arid climates with sand, high sunlight, and heat

exposure, requiring lightweight, portable, durable machines to facilitate increased field diagnostic accuracy [2, 9, 10]. POCUS can help users make urgent medical decisions, such as needle decompression for a pneumothorax or stabilizing an extremity with a fracture prior to transport [9, 10]. Similarly, MSK ultrasound can be used at high altitudes by hikers, climbers, and guides for long-bone fractures, and lung ultrasound can diagnosis pulmonary edema from low oxygen environments, expediting quick descent and medical treatment [70, 71].

POCUS use for trauma patients being evaluated by military personnel in combat zones or pre-hospital paramedics is useful to assess for pericardial or intra-abdominal free fluid, which assists in decisions to treat on scene or transport to a safer zone or hospital setting with higher level medical treatment capabilities [3–6]. In contrast, healthcare and military personnel working in low-resource, remote locations in very humid, hot, tropical settings such as the rainforest may evaluate patients with snakebites and other tropical diseases such as malaria or tuberculosis [59]. Outer space is an even more remote setting for POCUS use, where astronauts traveling and working in space shuttles or on the International Space Station can face unexpected problems such as biliary disease, nephrolithiasis, shoulder tendon tears, disc herniation, or other acute pathologies [26, 28, 68, 69]. Both outer space and certain combat zones can be further limited by confined spaces, reinforcing the need for small, portable, lightweight machines. Being surrounded by additional machinery and devices can create signal interference or problems with network connectivity if trying to transmit images to another location for review and image interpretation [2, 26, 28].

Finally, these studies have shown that the POCUS machines tested are durable in extreme temperature and ambient environmental settings, from extremely hot or humid climates to freezing temperatures on mountain tops at high altitudes and in outer space (also without gravity) without significant machine malfunctions or image degradation.

## Limitations

This was a narrative scoping review, thus the study is limited by the quality of the individual studies. Also, the study is limited by the search terms, as they may not be inclusive of all potential studies using POCUS in austere environments. We chose the inclusion and exclusion criteria to be broad, which required manual exclusion of a higher number of studies. This was done to capture as many relevant POCUS studies as possible to better inform readers and for planning of future clinical studies following the scoping review design. The study authors who reviewed all studies are trained in POCUS, which could introduce bias but would capture the most appropriate POCUS articles. RLS study reporting may be biased or limited based on places that have access to higher medical care or scientific support resources to publish in the literature.

The mixed numbers of high-quality literature such as RCTs, prospective trials, and meta-analysis studies in comparison to smaller prospective studies and case reports, and the variability in study quality per GRADE assessment, are limiting factors in using this data for future specific POCUS utilizations. Also, the many differences in study environment, geographic zones, and POCUS users makes it difficult to draw overall conclusions from these studies. However, we did find several systematic/scoping review papers (although some are now outdated from technological improvements over the past few years) and over 300 studies were found and described here. This review can serve as an outline with baseline data in designing and planning future research and clinical studies using POCUS in austere environments and RLS. We tried to highlight clinically important data and similarities and differences elucidated from reviewing these studies to guide future implementation of facilitators and overcoming barriers to current POCUS use in austere environments. Heterogeneity or meta-analysis were not performed due to the descriptive nature of the review.

## Future directions

With increased feasibility for POCUS outside of traditional expert use in the austere environment, caution must be used to ensure that the user receives appropriate device training to optimize diagnostic certainty and accuracy [1, 8]. Future studies can investigate the use of teleultrasound to transmit ultrasound images for remote review by ultrasound experts. Current studies have been published on testing of technical aspects of teleultrasound including image quality, transmission speeds, distances, and network types [8]. Most studies focus on diagnostic accuracy of teleultrasound in comparison to traditional in-person POCUS performance, and many have smaller sample sizes [8]. Thus, additional studies are needed to elucidate clinically relevant benefits and applications for teleultrasound in austere environments. Studies can also compare handheld versus portable laptop POCUS machines to determine if image quality is adequate for interpretation and if handheld machines have adequate battery support and durability in the field.

Finally, additional studies using artificial intelligence (AI) features to recognize abnormal pathology and assist in image interpretation for novice users would be useful. For example, studies are investigating automated B lines and calculation of cardiac ejection fraction in patients with pulmonary edema or acute heart failure, which could one day be self-performed by patients in their homes and transmitted for remote physician review using teleultrasound [73–75]. Other newer AI device functions can automatically calculate IVC volume as a measure of fluid status and bladder volume to evaluate for urinary retention to guide clinical management, especially in community hospital or RLS including LMIC [73–75]. Finally, AI functions are being incorporated into POCUS machine software as educational tools that can guide novice users or those in settings without access to expert reviewers to help obtain the proper POCUS views and alignments, particularly for cardiac views [73–75]. As technology continues to get faster and better with improved image quality, processing times, and transmissibility options, POCUS versatility and ability to improve medical care is anticipated to continue to grow in novel and innovative ways.

## Conclusion

Per our scoping review, POCUS use has been described in diverse global settings, including military and conflict zones, prehospital, RLS including LMIC, outer space, and high altitudes. POCUS has been used to assist user diagnostics and expedite medical treatment in patients with acute physical injuries or medical conditions [3]. The literature is heterogenous and of variable quality, with most studies done in prehospital or RLS including LMIC settings. Future high-quality studies are needed to further investigate the potential benefits of POCUS using teleultrasound, advanced imaging technologies, and smaller handheld devices to facilitate access and overcome current barriers in austere environments.

## Supporting information

**S1 Checklist. PRISMA-ScR guidelines for scoping reviews.**
(DOCX)

**S1 Appendix. Point-of-care ultrasound use in austere environments search terms.**
(DOCX)

**S1 Table. Summary of included ultrasound in austere environments studies in narrative scoping review.**
(DOCX)

**S2 Table. Austere ultrasound in prehospital and resource-limited settings summaries.**
(DOCX)

**S3 Table. Austere ultrasound included studies.**
(DOCX)

**S4 Table. Austere ultrasound excluded studies.**
(DOCX)

## Acknowledgments

Duke University medical library resources and librarian assistance with search terms and criteria.

## Author Contributions

**Conceptualization:** Rebecca G. Theophanous.

**Data curation:** Aubree Anderson, Rebecca G. Theophanous.

**Investigation:** Rebecca G. Theophanous.

**Methodology:** Aubree Anderson, Rebecca G. Theophanous.

**Writing – original draft:** Aubree Anderson, Rebecca G. Theophanous.

**Writing – review & editing:** Aubree Anderson, Rebecca G. Theophanous.

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
