## [Decision Letter · Decision Letter 0]

20 Sep 2024

PONE-D-24-38925Point-of-care ultrasound use in austere environments: a scoping reviewPLOS ONE

Dear Dr. Theophanous,

Thank you for submitting your manuscript to PLOS ONE. After careful consideration, we feel that it has merit but does not fully meet PLOS ONE’s publication criteria as it currently stands. Therefore, we invite you to submit a revised version of the manuscript that addresses the points raised during the review process.

We look forward to receiving your revised manuscript.

Kind regards,

Juan Antonio Valera-Calero

Academic Editor

PLOS ONE

2. As required by our policy on Data Availability, please ensure your manuscript or supplementary information includes the following:

Additional Editor Comments (if provided):

Reviewers' comments:

Reviewer's Responses to Questions

**Comments to the Author**

1. Is the manuscript technically sound, and do the data support the conclusions?

Reviewer #1: Yes

Reviewer #2: Yes

Reviewer #3: Yes

2. Has the statistical analysis been performed appropriately and rigorously? 

Reviewer #1: Yes

Reviewer #2: Yes

Reviewer #3: Yes

3. Have the authors made all data underlying the findings in their manuscript fully available?

Reviewer #1: Yes

Reviewer #2: Yes

Reviewer #3: Yes

4. Is the manuscript presented in an intelligible fashion and written in standard English?

Reviewer #1: Yes

Reviewer #2: Yes

Reviewer #3: Yes

5. Review Comments to the Author

Reviewer #1: Review Comments to the Author:

1.The manuscript presents a well-organized scoping review on the use of point-of-care ultrasound (POCUS) in austere environments. The methodology follows the PRISMA-ScR guidelines, and the GRADE framework is used to assess the quality of the included studies. However, I recommend changing the category of the manuscript from "Research Article" to "Review Article" to reflect its nature more accurately.

2. Statistical Analysis

Since this is a scoping review, statistical analysis is not the primary focus. The authors use descriptive statistics to summarize the number and types of studies included in the review, as well as their quality, as assessed by the GRADE tool. This approach is suitable for a review, but the manuscript does not include any advanced statistical analyses. Consider incorporating more advanced statistical tools where applicable to strengthen the findings.

3. Data Availability

The authors have made all relevant data underlying the manuscript's findings available without restriction, in accordance with PLOS ONE's data policy. This is clearly stated in the Data Availability Statement. As this is a review article, the data consist of extracted information from previously published studies, which are fully described in the manuscript and supporting information. The authors have met the journal's requirements by ensuring transparency in data sharing.

4. Language and Clarity

The manuscript is generally well-written and adheres to the standards of academic English. The scientific content is clear, but minor revisions could improve readability and flow. Specifically, some sentences are quite long and could be split into shorter, more concise statements. Additionally, there are a few grammatical issues and inconsistencies in the use of abbreviations that need to be addressed for better clarity. A thorough review for grammar and sentence structure would be beneficial.

5. General Comments

(A) This manuscript makes a valuable contribution to the understanding of POCUS applications in austere environments. It emphasizes the potential of portable ultrasound devices in resource-limited settings and other challenging environments. The use of the GRADE assessment tool adds rigor by providing an objective measure of the quality of the studies included, though it also highlights the need for more robust, high-quality research in this field.

(B) There are no concerns regarding dual publication, research ethics, or publication ethics. As this is a review article and does not involve new experimental research, participant consent and privacy issues are not applicable. The authors have declared no competing interests, and no specific funding was received, ensuring transparency.

(C) Other Recommendations for Revision:

1. Clarify and shorten complex sentences to improve readability.

2. Ensure consistent use of abbreviations after their introduction.

3. Correct minor grammatical and typographical errors noted during proofreading.

4. Expand the discussion on future research directions, particularly in relation to the use of AI and telemedicine in POCUS applications.

5. Consider using more statistical analysis tools where appropriate to provide deeper insights.

Reviewer #2: This manuscript by Theophanous RG et al. entitled “Point-of-care ultrasound（POCUS） use in austere environments: a scoping review” aims to explore the use of POCUS in a variety of Settings, including military and conflict areas, prehospital, RLS including LMIC, outer space, and high altitudes. The authors provide an interesting and potentially important manuscript that adequately describes current advances in the POCUS, as well as potential future directions. But there are still some content need to explain further.

Major：

1. What’s the difference between “point-of-care ultrasound” and “bedside ultrasound”? What is the reason for this paper to choose POCUS for in-depth study?

2. Because POCUS models and manufacturers are different, so the image display and application are different. Is there any relevant standard or classification basis in the process of literature collection? If you can add improved methods to the manuscript, it will be of great value to those who read the paper.

3. There were some minor issues with the references, including incomplete citations or outdated sources. Please check the references carefully and make the necessary corrections.

Reviewer #3: International work! The manuscript entitled "Point-of-care ultrasound use in austere environments: a scoping review” updated narrative scoping review describes POCUS use in these global settings. The novelty factor of the article is good. I find the article potentially acceptable for publication after minor revision, my comments are below:

1. Please standardize the font, including the documents in the supplementary materials.

2. Please add tables or figures to summarize each setting.

3. Please unify the tables into three-line tables.

6. PLOS authors have the option to publish the peer review history of their article (what does this mean?). If published, this will include your full peer review and any attached files.

Reviewer #1: No

Reviewer #2: No

Reviewer #3: No

---

## [Author Response · Author response to Decision Letter 0]

23 Sep 2024

Responses to reviewers and editors are included in the attached files. Thank you for your timely review and very useful comments. We incorporated all suggested feedback as suggested to meet the journal's requirements and strengthen the manuscript.

---

## [Decision Letter · Decision Letter 1]

30 Sep 2024

Point-of-care ultrasound use in austere environments: a scoping review

PONE-D-24-38925R1

Dear Dr. Theophanous,

We’re pleased to inform you that your manuscript has been judged scientifically suitable for publication and will be formally accepted for publication once it meets all outstanding technical requirements.

Kind regards,

Juan Antonio Valera-Calero

Academic Editor

PLOS ONE

Additional Editor Comments (optional):

Reviewers' comments:

Reviewer's Responses to Questions

**Comments to the Author**

1. If the authors have adequately addressed your comments raised in a previous round of review and you feel that this manuscript is now acceptable for publication, you may indicate that here to bypass the “Comments to the Author” section, enter your conflict of interest statement in the “Confidential to Editor” section, and submit your "Accept" recommendation.

Reviewer #1: All comments have been addressed

Reviewer #2: All comments have been addressed

Reviewer #3: All comments have been addressed

2. Is the manuscript technically sound, and do the data support the conclusions?

Reviewer #1: Partly

Reviewer #2: Yes

Reviewer #3: Yes

3. Has the statistical analysis been performed appropriately and rigorously? 

Reviewer #1: No

Reviewer #2: Yes

Reviewer #3: N/A

4. Have the authors made all data underlying the findings in their manuscript fully available?

Reviewer #1: Yes

Reviewer #2: Yes

Reviewer #3: Yes

5. Is the manuscript presented in an intelligible fashion and written in standard English?

Reviewer #1: Yes

Reviewer #2: Yes

Reviewer #3: Yes

6. Review Comments to the Author

Reviewer #1: In this review, the authors aim to identify the applications of POCUS in austere environments. The usability of POCUS in such settings is found to be generally positive. Additionally, the authors have summarized the various applications of POCUS, highlighting its benefits in austere environments.

The future directions of the research focus on advancing teleultrasound, integrating AI, improving training tools, testing device durability, and expanding POCUS applications in resource-limited and extreme settings. These efforts are expected to enhance the accessibility, accuracy, and overall impact of POCUS in challenging global contexts.

Main Suggestions

The authors have used the GRADE system to evaluate the quality of the articles included in this review. To provide more clarity, I suggest that the authors describe the process of applying GRADE in detail in the revised manuscript.

The GRADE (Grading of Recommendations, Assessment, Development, and Evaluations) tool is a widely used system for assessing the quality of evidence in systematic reviews, meta-analyses, and other research studies. It involves evaluating the certainty of the evidence and the strength of the recommendations based on several predefined criteria.

Reviewer #2: (No Response)

Reviewer #3: The manuscript entitled "Point-of-care ultrasound use in austere environments: a scoping review” updated narrative scoping review describes POCUS use in these global settings. The paper was well structured, and the novelty factor of the paper was generally good. The authors have well completed the revision. In my opinion, this paper is suitable to be published in PLOS ONE.

7. PLOS authors have the option to publish the peer review history of their article (what does this mean?). If published, this will include your full peer review and any attached files.

Reviewer #1: No

Reviewer #2: No

Reviewer #3: No

---

## [Editor Report · Acceptance letter]

10 Oct 2024

PONE-D-24-38925R1 

PLOS ONE

Dear Dr. Theophanous, 

I'm pleased to inform you that your manuscript has been deemed suitable for publication in PLOS ONE. Congratulations! Your manuscript is now being handed over to our production team.

Kind regards, 

on behalf of

Dr. Juan Antonio Valera-Calero 

Academic Editor

PLOS ONE